# A comparison of drying methods on the quality for bryophyte molecular specimens collected in the field

**Fengjiao Shen**[1◉], **Lin Li**[1◉], **Dan Wang**[1], **Mengzhen Wang**[1], **James R. Shevock**[2], **Jiancheng Zhao**[1], **Shuo Shi**[1]*

**1** College of Life Sciences, Hebei Normal University, Shijiazhuang, Hebei, China, **2** Department of Botany, California Academy of Sciences, San Francisco, California, United States of America

◉ These authors contributed equally to this work.
* shishuo@hebtu.edu.cn

**Data Availability Statement:** All relevant data are within the paper and its Supporting information files.

## Abstract

A major challenge in extracting high-quality DNA from bryophytes is the treatment of bryophyte material in the field. The existing and commonly used treatment methods in the field have several shortcomings. Natural drying methods can lead to DNA breaks. In addition, it is highly cumbersome to carry large quantities of silica gel in the field due to its weight and high risk of contamination among samples. In this study, we explored more convenient drying methods to treat bryophyte specimens and promote more efficient DNA recovery. The quantity and quality of genomic DNA extracted from every bryophyte species using different drying methods, including hot-air drying methods (150˚C, 80˚C, and 40˚C), natural drying method, and silica gel drying method, were measured. Spectrophotometry, electrophoresis, and PCR amplification were performed to assess the effects of different drying methods. The results of total DNA purity, total DNA concentration, PCR success, and OD 260/230 ratios suggested that the hot-air drying (40–80˚C) was the best method. The morphological comparison revealed that hot-air drying at 40˚C and 80˚C exerted no significant adverse effects on plant morphology and taxonomic studies. Thus, this method prevents rapid DNA degradation and silica gel pollution and saves the workforce from carrying large amounts of silica gel to the field. Several inexpensive devices, such as portable hairdryers, fan heaters, and electric blankets, are available that can be easily carried to the field for drying molecular specimens.

## Introduction

The drying treatment methods of bryophyte specimens greatly influence the quality of DNA while collecting specimens in the field [1, 2]. Unlike seed plant collection, traditional treatment of the bryophyte specimens does not include immediate drying in the field. Bryophyte specimens have to be dried naturally for several days; however, the natural drying treatment can cause bryophyte DNA breaks [1, 2]. It is therefore always difficult to obtain pure DNA and a

**Funding:** The authors report the following sources of funding: National Natural Science Foundation of China [31370237 and 31370184], Youth Foundation of Education Department of Hebei Province, No. QN2017342, http://jyt.hebei.gov.cn/, awarded to SS, Natural Science Foundation of Hebei Province under Grant, No. C2019205175, https://kjt.hebei.gov.cn/jjb/, awarded to LL, Innovation Fund Project for Graduate Student of Hebei Province, No. CXZZBS2019088, http://jyt.hebei.gov.cn/, awarded to FS, and Innovation Funding Program for Graduate Students of Hebei Normal University, No. CXZZSS2021066, http://www.hebtu.edu.cn/. awarded to FS. The funders had no role in study design, data collection, and analysis, decision to publish, or preparation of the manuscript.

**Competing interests:** NO authors have competing interests Enter: The authors have declared that no competing interests exist.

high percentage of PCR from the bryophyte specimens used for molecular experiments stored in herbaria or natural history museums. The ideal treatment method used for plant molecular specimens (especially for DNA extraction from plant leaves) is the low temperature (e.g., by −80˚C refrigeration, −196˚C liquid nitrogen, or dry ice). However, such methods are inconvenient to be implemented in the field [3–5]. Although NaCl/CTAB (Cetyltrimethylammonium Bromide) solution can be used for fieldwork [6], it is also not convenient to carry liquids in the field, especially for large collections. Doyle and Dickson [7] found that the dried molecular specimen could be used for DNA extraction. In addition, it could be easily stored and carried, suggesting drying as one of the most appropriate methods in the field.

Although there were some case studies showed the DNA degrades during the drying process [8, 9], different drying methods and techniques have been applied to several groups of organisms, such as plants, animals, and macrofungi, to obtain molecular data in the last 30 years. Some of these methods include the alcohol method [10], silica gel method [11], diatomite and sand burying method [12], and physical and chemical desiccation [4, 13]. At present, silica gel drying is mostly used for collecting molecular specimens in the field [12, 14–18]. However, the silica gel drying method has certain limitations. First, it is inconvenient to carry silica gel in large quantities to the field, especially in remote field collection areas. Second, failure to replace silica gel following water absorption results in incomplete drying of the specimen, adversely affecting the quality of a molecular specimen [6]. Moreover, the repeated replacement of water-absorbing silica gel to ensure samples dry quickly and completely will drastically increase the fieldwork time, especially when collecting in humid environments. Third, the recycling treatment can result in cross-contamination among specimens despite specific molecular specimen bags assigned to each molecular specimen (the bag is synthesized from a breathable non-woven fabric). In conclusion, the field specimen collection step using silica gel suffers from several shortcomings during the processing of a bryophyte molecular specimen.

Considering the above problems, the hot-air drying method has emerged as a method of choice for bryophyte molecular specimens in the field. The preliminary experiment on seed plants and macrofungi showed that the high-quality DNA could be efficiently obtained from the specimen subjected to hot-air drying [19–21]. Therefore, we studied the effects of different drying methods on the quality of extracted DNA and its suitability for PCR.

Another problem encountered during bryophyte molecular specimen collection is that bryophytes are small plants and several individuals of different species can grow together as mixed populations. This makes it difficult to separate different bryophyte species properly and timely in the field. It is difficult to ensure that the molecular specimens of all species are collected in the field, if a part of the specimen is taken as a molecular specimen, as is the case with angiosperms. However, this problem could be overcome if all individuals in a bryophyte specimen were treated as one molecular sample by the hot-air drying method. Species could be separated later in the laboratory for DNA extraction and other studies. Therefore, in this study, the bryophyte specimen quality was evaluated after treated by hot-air drying method and other methods.

## Materials and methods

### Materials

For the study, four species were selected as different types of bryophytes based on plant size, branching pattern, and leaf texture. These included three mosses, namely *Campylopus schmidii* (Müll. Hal.) A. Jaeger, *Polytrichum commune* Hedw., *Hypnum calcicola* Ando, and one liverwort, *Marchantia polymorpha* L. All four samples were kept fresh at the beginning of the

experiment. The information of voucher specimens is shown in Table 1. The voucher specimens were deposited in the herbarium of the Hebei Normal University (HBNU).

## Methods

**Material processing.** Five drying methods, namely the hot-air drying method (150°C, 80°C, and 40°C), silica gel drying method, and natural drying method, were compared to treat bryophyte molecular specimens. The specific methods were:

i. In order to ensure the stability of the experimental environment and equipment, the materials were dried in the laboratory in this experiment. Before entering the laboratory, we placed the material in sealed plastic bag with small holes to keep the samples living. When the bryophyte samples arrived at the laboratory, the soil on four fresh bryophyte samples were cleaned with water. And then, the water on the bryophyte surface was sucked up by the absorbent paper.

ii. The material of each specimen was divided into nine parts, and each part was placed in molecular specimen bags. Three parts were used for pre-experiment, and six parts were used for the formal experiment. The fresh weight of each part of the specimen was ≥200 mg.

iii. In the pre-experiment, the three parts materials were placed in an electric thermostatic drying oven (DHG-9240A, Zhongyiguoke Tech. Beijing, China) at 150°C, 80°C, and 40°C, respectively, until the weight of bryophytes reached a constant value. The device can control the temperature and function like an air blast. The rate of water loss was calculated, and the time required at each temperature was recorded.

iv. In the formal experiment, there are six parts materials for the experiment. These three parts were placed in the oven electric thermostatic drying at 150°C, 80°C, and 40°C, respectively (the drying time were the same as pre-experiment). The fourth part material was kept in a paper bag in a cool, well-ventilated place. The fifth part material was collected in a sealed plastic bag with excess dry silica gel. The sixth part material was contained in a sealed plastic bag and placed in a –80°C refrigerator. After all of the samples, except the sixth ones, were dry, the follow-up experiment was performed.

## DNA extraction and quality examination

i. DNA extraction
The experimental materials of five different drying methods, with 16 repeats and 5 mg of each, were weighed. A high-throughput tissue grinding mill (SCIENTZ-48, Ningbo Xinzhi Biotechnology Co., LTD, Ningbo, China) was used to quickly grind the specimen into powder. The mCTAB method was used to extract DNA [22].

**Table 1. The information of voucher specimens.**

| Species | Specimen No. | Collectors | Collection sites | Families |
|---|---|---|---|---|
| *Campylopus schmidii* A. Jaeger | SFJTX003 | Shen et al. | Yunnan, China | Dicranaceae |
| *Polytrichum commune* Hedw. | 20166223 | Niu | Guizhou, China | Polytrichaceae |
| *Hypnum calcicola* Ando | 201662225 | Duan | Guizhou, China | Hypnaceae |
| *Marchantia polymorpha* L. | SFJTX004 | Shen | Hebei, China | Marchantiaceae |

ii. DNA examination

The quality of DNA was assessed using a micro-spectrophotometer (NanoDrop 2000) and agarose gel electrophoresis. Because the absorption at 230 nm can be caused by small organic compounds, we recorded the OD 260/230 ratio. The DNA concentration was used to assess the purity and amount of DNA obtained. The fragment size, degradation, and concentration of DNA were checked using 1% agarose gel electrophoresis. ImageJ (v1.4.3.67) software was used to conduct quantitative analysis on high-molecular weight genomic DNA (which is close to gel hole position in the gel image) in the total DNA agarose gel electrophoretogram results.

STATISTICA (v10.0.228.8) was used to conduct statistical analysis of the data obtained from micro-spectrophotometer spectrophotometry (OD 260/230 ratio, total DNA concentration) and agarose gel electrophoretogram (the concentration of high-molecular weight genomic DNA). The results were analyzed using the letter-marking multiple comparison method [23] (S1 and S2 Tables).

iii. DNA quality examination by PCR

To a certain extent, the quality of DNA can be reflected by the success rate of PCR amplification. The presence of an increased number of high-molecular weight genomic DNA is associated with a high amplification success rate. The PCR primers ITS-P5 "5'–3', CCTTATCAYTTAGAGGAAGGAG" and ITS-U4 "5'–3', RGTTTCTTTTCCTCCGCTTA" were used, which are designed for plants. The PCR amplification procedure reference to Cheng et al. [24], and the annealing temperature is 55°C. PCR products were checked using 1% agarose gel electrophoresis. The success rate of PCR was calculated to evaluate the quality of molecular samples.

**Comparison of morphological characteristics.** Both macroscopic and microscopic morphological characteristics are the important evidence for bryophyte species identification. It is unknown if the treatments used in this study affected the morphological identification of the specimen. Therefore, the overall plant morphology, leaf characteristics, and leaf transverse section characteristics of a single specimen after different drying methods were compared with those of the traditional naturally dried samples. If these characteristics were consistent, it indicated that different drying methods did not affect the morphological identification of the specimens.

## Results

### DNA quality analysis

**Examination of DNA purity.** The OD 260/230 ratios of four bryophytes were compared (Fig 1; S1 Table).

Except for the fresh freezing control group, consisting of *P. commune* and *H. calcicola*, the DNA OD 260/230 ratios obtained after the hot-air drying method at 80°C were the highest. The DNA OD 260/230 ratios obtained after the silica gel drying method were the lowest. For *M. polymorpha*, the DNA OD 260/230 ratios obtained following the hot-air drying method at 80°C and 40°C and the natural drying method were the highest. For *C. schmidii*, DNA OD 260/230 ratios obtained after different drying methods showed no significant difference ($p > 0.05$).

**Examination of DNA concentration.** The DNA concentrations of four bryophytes examined by a microspectrophotometer (total DNA concentration, Fig 2) and agarose gel electrophoretogram (high-molecular weight genomic DNA) were compared (Fig 3, S1–S4 Figs).

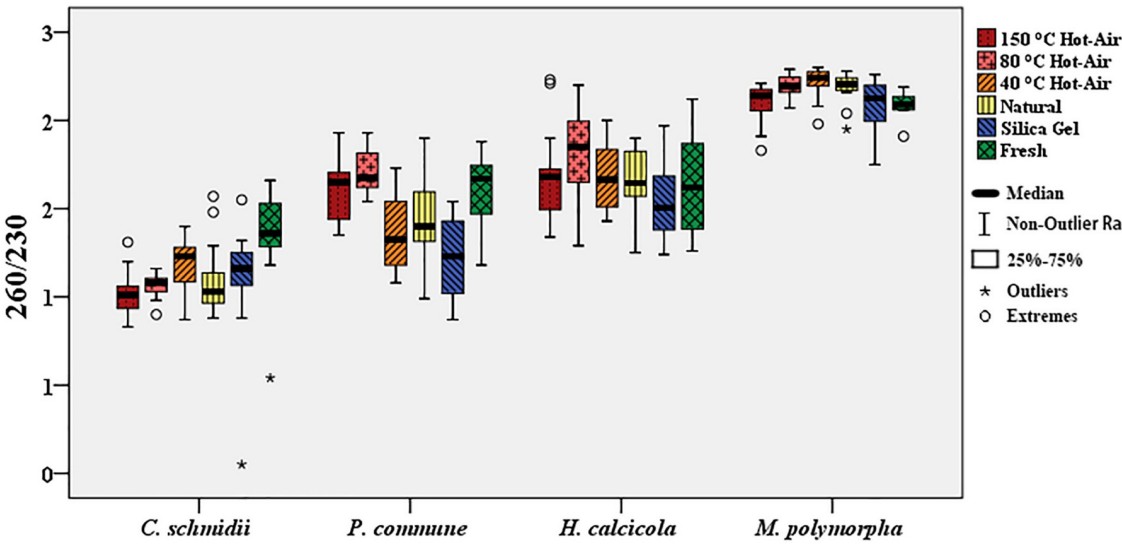

**Fig 1. The extracted DNA purity after different drying methods (OD 260/230).**

For *P. commune*, except for the fresh freezing control group, the total DNA concentration obtained after hot-air drying at 80˚C was the highest. For *H. calcicola*, the total DNA concentration obtained after hot-air drying at 40˚C, 80˚C, natural drying, and silica gel drying resulted in an insignificant difference ($p > 0.05$). The total DNA concentrations obtained after these methods were higher than that obtained with hot-air drying at 150˚C. For *C. schmidii*, the total DNA concentration obtained after different drying methods had no significant difference ($p > 0.05$). For *M. polymorpha*, the total DNA concentration of the five treatments had no significant difference ($p > 0.05$).

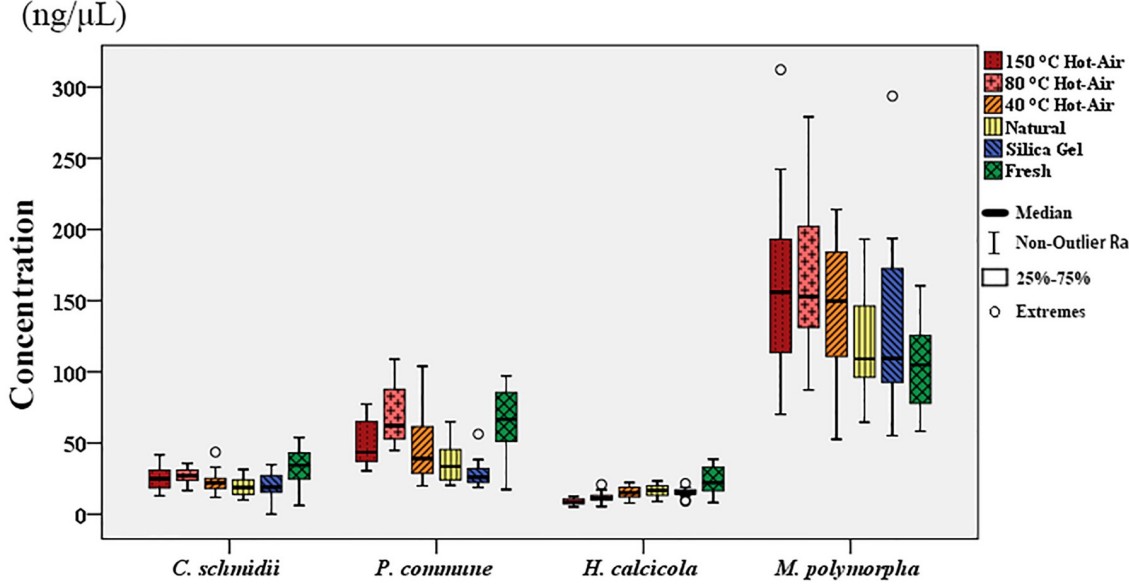

**Fig 2. The concentration of extracted total DNA after different drying methods was detected by a microspectrophotometer.**

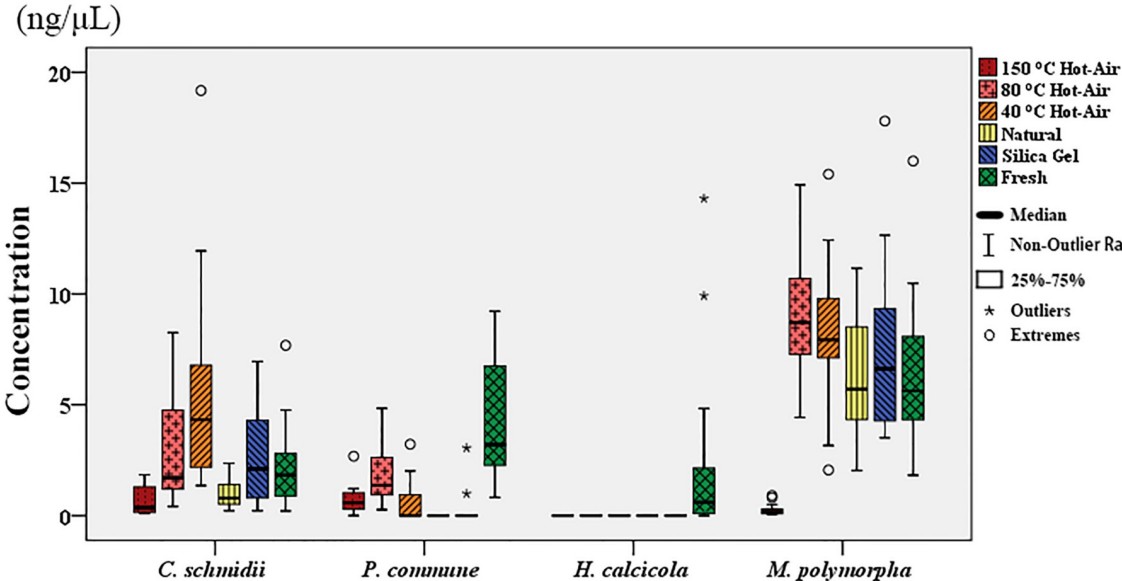

**Fig 3. The concentration of extracted high-molecular weight genomic DNA by agarose gel electrophoretogram after different drying methods.**

Electrophoresis results for high-molecular weight genomic DNA showed that for *P. commune*, except the fresh freezing control group, the concentration of high-molecular weight genomic DNA obtained after hot-air drying at 80˚C was the highest. For *C. schmidii*, the concentration of high-molecular weight genomic DNA obtained after hot-air drying at 40˚C was the highest. For *M. polymorpha*, only the concentration of high-molecular weight genomic DNA obtained after 150˚C is the lowest. The concentration of high-molecular weight genomic DNA of *H. calcicola* received after different drying methods had no significant difference (p > 0.05) (Fig 3).

## PCR amplification products

The PCR amplification for samples of five different methods and the control group was conducted (S5–S8 Figs). The statistics were obtained for assessing the success rate of PCR amplification, and the results are shown in Fig 4. For the four bryophytes, there was no statistically significant difference in PCR success rate of different drying methods. However, the amplification rate of the four samples after hot-air drying at 80˚C and 40˚C was higher. The amplification rate of the four samples after hot-air drying at 80˚C was 100%, and the amplification rate of the three samples after hot-air drying at 40˚C was 100%. The success rate of PCR amplification was slightly lower after hot-air drying at 40˚C; however, it was higher than that obtained from other methods.

## Morphological comparison before and after drying

The overall plant morphology, leaf characteristics, and leaf transverse section characteristics (S9–S11 Figs) of four samples before and after different drying treatments were compared. Hot-drying at 40˚C and 80˚C, silica gel, and natural drying treatments resulted in insignificant differences in the morphology characteristics, especially regarding the major characteristics of identification, such as plant color, leaf morphology, the shape of the cell, and transverse section characteristics of molecular specimens. The only exception was the material dried at 150˚C,

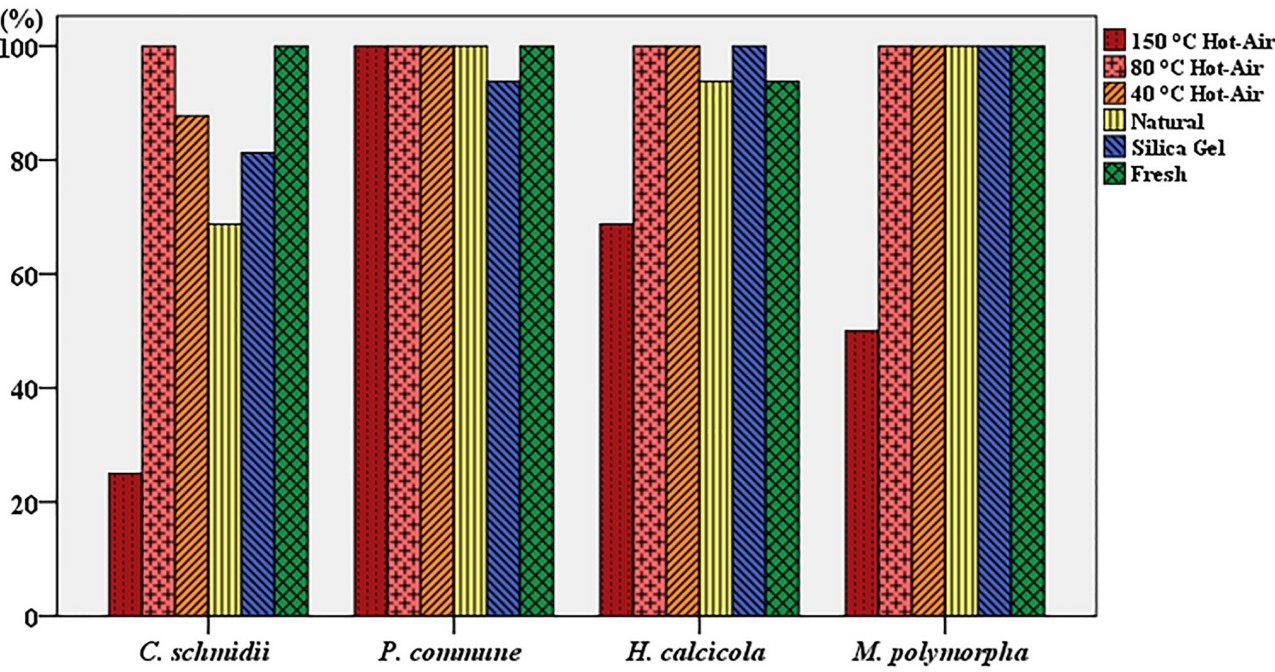

**Fig 4. Success rates of PCR amplification.**

which was darker in color. It can be concluded that hot drying at 40°C and 80°C, silica gel, and natural drying treatments do not affect the identification of bryophytes.

## Discussion

The results showed that the effect of the hot-air drying method at 40°C and 80°C was better than that of the silica gel drying method and natural drying method. In addition, the silica gel drying method was inconvenient to perform in the field, and the natural drying method was highly affected by environmental humidity. Therefore, 40°C to 80°C hot-air drying method for specimen drying is recommended in the field to avoid rapid degradation of DNA. More-over, this method does not damage the characteristics of the traditional morphological study. The field collection can be created in different ways, such as an electric blanket, hairdryer, or a portable fan heater. In this study, three temperatures were selected to form a temperature gradient. Among these, 150°C, 80°C, and 40°C were simulation temperatures corresponding to different distances from the vent of the portable fan heater.

In addition to drying methods, plants should begin the drying process as soon as possible after the field collection event. The best practice is to begin the drying process on the day of collection. The following points should be noted during the specific implementation of the operation: 1) Specimens stored in open packets can offer more airflow, especially if the packets are placed upright to aid drying. 2) If the sample is wet, for example, collected from water, it should be carefully processed during hot drying to keep a lower temperature. The samples should be squeezed first to release most of the excess water. 3) Because high temperature and humidity can damage the molecular material, large quantities of materials should not be placed in the same sample bag or packet.

Interestingly, the results of our study resemble those obtained for angiosperms [20] and mushrooms [21]. In these two studies, temperatures used for molecular specimen drying were

40˚C and 70˚C, respectively. And cut into pieces should be needed to reduce the drying time and improve the molecular specimen quality. At present, lichen specimens, similar to bryophyte specimens, are dried primarily by the natural drying method. It is challenging to extract DNA from lichen specimens, which dried naturally, after being collected for two years [25]. Moreover, there exist several other limitations in treating animal molecular specimens [26]. The results of the drying method in this study and the above research provide a reference for the treatment of other organism DNA specimens, e.g., algae, lichen, and animals.

## Conclusions

It is demonstrated in this study that the hot-air drying (40–80˚C) offered the best results for drying bryophyte molecular specimens as soon possible after a collecting event. This method causes little damage to the DNA in bryophyte samples and is also convenient to operate. It is recommended that this method be used in the future for drying bryophyte specimens in the field.

## Supporting information

**S1 Table. Comparison of OD 260/230 values of the four bryophytes' DNA after different drying treatments.** 150, 150˚C hot-air drying; 80, 80˚C hot-air drying; 40, 40˚C hot-air drying; N, natural drying; S, silica gel drying; F, fresh sample; [a,b,c,d] The superscript of same letters indicate that there is no statistically significant difference (P>0.05), the superscript of different letters indicate that there is a statistically significant difference (P<0.05).
(DOCX)

**S2 Table. Comparisons of extract DNA concentrations of the four bryophytes after different drying treatments.** 150, 150˚C hot-air drying; 80, 80˚C hot-air drying; 40, 40˚C hot-air drying; N, natural drying; S, silica gel drying; F, fresh sample; DNA-N means total DNA concentration form Nanodrop 2000 micro-spectrophotometer and DNA-G means long fragment DNA from agarose gel electrophoretogram. [a,b,c,d] The superscripts of the same letters indicate that there is no statistically significant difference (P>0.05). The superscripts of the different letters indicate that there is a statistically significant difference (P<0.05) $^{-1}$.
(DOCX)

**S1 Fig. Agarose gel electrophoresis of Genomic DNA of *C. schmidii* obtained from different drying methods.** Note, 150, 150˚C hot-air drying; 80, 80˚C hot-air drying; 40, 40˚C hot-air drying; N, natural drying; S, silica gel drying.
(TIF)

**S2 Fig. Agarose gel electrophoresis of Genomic DNA of *P. commune* obtained from different drying methods.** Note, 150, 150˚C hot-air drying; 80, 80˚C hot-air drying; 40, 40˚C hot-air drying; N, natural drying; S, silica gel drying.
(TIF)

**S3 Fig. Agarose gel electrophoresis of Genomic DNA of *H. calcicola* obtained from different drying methods.** Note, 150, 150˚C hot-air drying; 80, 80˚C hot-air drying; 40, 40˚C hot-air drying; N, natural drying; S, silica gel drying.
(TIF)

**S4 Fig. Agarose gel electrophoresis of Genomic DNA of *M. polymorpha* obtained from different drying methods.** Note, 150, 150˚C hot-air drying; 80, 80˚C hot-air drying; 40, 40˚C hot-air drying; N, natural drying; S, silica gel drying.
(TIF)

**S5 Fig. Agarose gel electrophoresis of PCR products of *C. schmidii* obtained from different drying methods.** Note, 150, 150˚C hot-air drying; 80, 80˚C hot-air drying; 40, 40˚C hot-air drying; N, natural drying; S, silica gel drying.
(TIF)

**S6 Fig. Agarose gel electrophoresis of PCR products of *P. commune* obtained from different drying methods.** Note, 150, 150˚C hot-air drying; 80, 80˚C hot-air drying; 40, 40˚C hot-air drying; N, natural drying; S, silica gel drying.
(TIF)

**S7 Fig. Agarose gel electrophoresis of PCR products of *H.calcicola* obtained from different drying methods.** Note, 150, 150˚C hot-air drying; 80, 80˚C hot-air drying; 40, 40˚C hot-air drying; N, natural drying; S, silica gel drying.
(TIF)

**S8 Fig. Agarose gel electrophoresis of PCR products of *M. polymorpha* obtained from different drying methods.** Note, 150, 150˚C hot-air drying; 80, 80˚C hot-air drying; 40, 40˚C hot-air drying; N, natural drying; S, silica gel drying.
(TIF)

**S9 Fig. The morphological characters of overall plants.** Note, C, *C. schmidii*; P, *P. commune*; H, *H. calcicola*; M, *M. polymorpha*; 150, 150˚C hot-air drying; 80, 80˚C hot-air drying; 40, 40˚C hot-air drying; N, natural drying; S, silica gel drying. Bar scales C/P/H = 1 mm; M = 1 cm.
(TIFF)

**S10 Fig. The morphological characters of leaves.** Note, C, *C. schmidii*; P, *P. commune*; H, *H. calcicola*; M, *M. polymorpha*; 150, 150˚C hot-air drying; 80, 80˚C hot-air drying; 40, 40˚C hot-air drying; N, natural drying; S, silica gel drying. Bar scales C/P = 0.5 mm; H = 0.1 mm; M = 1 mm.
(TIFF)

**S11 Fig. The morphological characters of transverse sections.** Note, C, *C. schmidii*; P, *P. commune*; H, *H. calcicola*; M, *M. polymorpha*; 150, 150˚C hot-air drying; 80, 80˚C hot-air drying; 40, 40˚C hot-air drying; N, natural drying; S, silica gel drying. Bar scales = 50 μm.
(TIFF)

**S1 Raw images.**
(PDF)

## Acknowledgments

Thanks to Dr. Jing-Yuan Niu and Dr. Xu-Hong Duan for collecting the specimens; Dr. Hai-Yan Liu, for assistance with statistical analysis; Dr. Shi-Liang Zhou, for his helpful advice and revision of the draft. We thank TopEdit (www.topeditsci.com) for linguistic assistance during the preparation of this manuscript.

## Author Contributions

**Conceptualization:** Lin Li, Shuo Shi.

**Data curation:** Fengjiao Shen.

**Formal analysis:** Fengjiao Shen, Shuo Shi.

**Funding acquisition:** Fengjiao Shen, Lin Li, Shuo Shi.

**Investigation:** Fengjiao Shen, Dan Wang, Mengzhen Wang.

**Methodology:** Fengjiao Shen, Lin Li, Jiancheng Zhao, Shuo Shi.

**Project administration:** Fengjiao Shen, Lin Li, Shuo Shi.

**Software:** Fengjiao Shen.

**Supervision:** Shuo Shi.

**Validation:** Fengjiao Shen, Lin Li.

**Visualization:** Fengjiao Shen, Shuo Shi.

**Writing – original draft:** Fengjiao Shen, Lin Li, James R. Shevock, Jiancheng Zhao, Shuo Shi.

**Writing – review & editing:** Fengjiao Shen, Lin Li, James R. Shevock, Shuo Shi.

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
