## [Decision Letter · Decision Letter 0]

29 Aug 2022

PONE-D-22-14636A comparison of drying methods on the quality for bryophyte molecular specimens collected in the fieldPLOS ONE

Dear Dr. Shuo Shi

Thank you for submitting your manuscript to PLOS ONE. After careful consideration, we feel that it has merit but does not fully meet PLOS ONE’s publication criteria as it currently stands. Therefore, we invite you to submit a revised version of the manuscript that addresses the points raised during the review process.

We look forward to receiving your revised manuscript.

Kind regards,

Rosani do Carmo de Oliveira Arruda

Academic Editor

PLOS ONE

https://journals.plos.org/plosone/s/file?id=ba62/PLOSOne_formatting_sample_title_authors_affiliations.pdf".

“SS, No. QN2017342, Youth Foundation of Education Department of Hebei Province, http://jyt.hebei.gov.cn/, No

LL, No. C2019205175, Natural Science Foundation of Hebei Province under Grant, https://kjt.hebei.gov.cn/jjb/, No

FS, No. CXZZBS2019088, Innovation Fund Project for Graduate Student of Hebei Province, http://jyt.hebei.gov.cn/, No

LS, No. CXZZSS2021066, Innovation Funding Program for Graduate Students of Hebei Normal University, http://www.hebtu.edu.cn/, No”

Please state what role the funders took in the study.  If the funders had no role, please state: ""The funders had no role in study design, data collection and analysis, decision to publish, or preparation of the manuscript.

3. PLOS requires an ORCID iD for the corresponding author in Editorial Manager on papers submitted after December 6th, 2016. Please ensure that you have an ORCID iD and that it is validated in Editorial Manager. To do this, go to ‘Update my Information’ (in the upper left-hand corner of the main menu), and click on the Fetch/Validate link next to the ORCID field. This will take you to the ORCID site and allow you to create a new iD or authenticate a pre-existing iD in Editorial Manager. Please see the following video for instructions on linking an ORCID iD to your Editorial Manager account: https://www.youtube.com/watch?v=_xcclfuvtxQ.

Additional Editor Comments:

REVIEWER 1.

Title: 1. The title of manuscript is little inappropriate and incomplete.

Introduction

The text should be more inclined towards the problems associated with the sampling and DNA isolation from bryophytes species. What are the methods traditionally used for the collection of bryophytes sample? Why different treatment is required and what are their drawbacks?

Materials and Methods

1. Line 76-77: What is the location or collection site of the studied bryophytes specimen?

2. Line 79-80: All four samples were kept fresh at the beginning of the experiment…??? Means stored in -80oC

3. Line 86: The collected bryophytes samples from field contains mud, dirt and microscopic organisms which may cause undesirable PCR amplification. What are the steps taken to clean the collected samples?

4. Line 94-100: Some parts should be reconsidered and rewritten. Experiment design is not clear from the text. From the text it appears that samples are dried in oven at 150°C, 80°C, and 40°C, respectively and then kept in a paper bag in a cool, well-ventilated place, second in a sealed plastic bag with excess silica gel, and third in a sealed plastic bag in a –80°C refrigerator and then again dried at 150°C, 80°C, and 40°C. That is bit confusing.

5. Line 121: Please specify the sequence and Tm value of PCR primers ITS-P5 and ITS-U4 used in present study.

6. Line 124-127: Please specify the method used for evaluation of effect of different drying method on the morphological characteristics of bryophytes.

7. What concentration of gel is used for the gel electrophoresis (DNA isolation and PCR products)??

Results:

1. Line 131: The ratio of the spectrophotometric absorbance of sample at 260/280 and 260/230 are generally used to determine the purity of nucleic acid contaminated by phenol or proteins and organic acids respectively. The present study includes DNA purity result based on only the 260/230 ratio.

2. S1 Table: Title should be completed, must be well understandable without the text. What is the difference between DNA purity and DNA concentration (Table 1& 2)? Please provide details of abbreviations (C, P, H, M, N, S, F) in the footnote of table.

3. Figure (1-9): Legends should be completed and well understandable without text.

4. Line 133-138 (Table 1): Spectrophotometer, nanodrops and gel electrophoretogram are used to determine the concentration of DNA. The extracted DNA having a ratio of absorbance A260/280- 1.8 and A260/230- 2.0-2.2 are considered as pure. The A260/230 values of three samples are less than 2.0. The contamination of organic acids might be present in sample. Please provide the justification of statement in your manuscript

5. Instead of long fragment DNA author can use “high-molecular weight genomic DNA”.

6. Provide details (name, collection site, specimen no. family, etc.,)in tabular form about the bryophytes specimens used in present study.

Discussion

Please cite the literature associated with problems arises during collection storage and DNA isolation from bryophyte specimens and how this study provides solutions to those problems. Please give conclusion and future prospects of your study.

REVIEWER 2.

The manuscript brings a simple but important technical advance in the area of molecular studies of bryophytes. Considering the findings presented here, it is possible to carry out the field collection more easily and, at the same time, guarantee the obtaining of quality DNA samples for molecular investigations.

Some minor revisions are necessary:

1- Page 5, line 102: write only "DNA extraction"

2- Page 8, lines 147 to 152: for H. calcicola, the results for hot-air drying at 40°C was not commented. The results obtained for M. polymorpha was not commented. Need to reference that the paragraph is the analysis of Fig. 2.

3- Page 8, lines 153 to 155: consider reviewing the comparison between the results for the fresh freezing control group and hot-air drying at 80°C for M. polymorpha; in table S2 it is clear that, for this species, the only treatment that showed a significant difference (p > 0.05) was drying with hot air at 150°C.

4- Page 8, lines 161 to 163: consider reviewing the comparison of the success rate of PCR amplification after hot-air drying at 80°C and the other treatments; in Fig. 4 (and supplementary figures as well) it is not possible to detect a difference between this treatment and others in some cases.

Reviewers' comments:

Reviewer's Responses to Questions

**Comments to the Author**

1. Is the manuscript technically sound, and do the data support the conclusions?

Reviewer #1: Partly

Reviewer #2: Yes

2. Has the statistical analysis been performed appropriately and rigorously? 

Reviewer #1: Yes

Reviewer #2: Yes

3. Have the authors made all data underlying the findings in their manuscript fully available?

Reviewer #1: Yes

Reviewer #2: Yes

4. Is the manuscript presented in an intelligible fashion and written in standard English?

Reviewer #1: No

Reviewer #2: Yes

5. Review Comments to the Author

Reviewer #1: Title:

1. The title of manuscript is little inappropriate and incomplete.

Introduction

The text should be more inclined towards the problems associated with the sampling and DNA isolation from bryophytes species. What are the methods traditionally used for the collection of bryophytes sample? Why different treatment is required and what are their drawbacks?

Materials and Methods

1. Line 76-77: What is the location or collection site of the studied bryophytes specimen?

2. Line 79-80: All four samples were kept fresh at the beginning of the experiment…??? Means stored in -80oC

3. Line 86: The collected bryophytes samples from field contains mud, dirt and microscopic organisms which may cause undesirable PCR amplification. What are the steps taken to clean the collected samples?

4. Line 94-100: Some parts should be reconsidered and rewritten. Experiment design is not clear from the text. From the text it appears that samples are dried in oven at 150°C, 80°C, and 40°C, respectively and then kept in a paper bag in a cool, well-ventilated place, second in a sealed plastic bag with excess silica gel, and third in a sealed plastic bag in a –80°C refrigerator and then again dried at 150°C, 80°C, and 40°C. That is bit confusing.

5. Line 121: Please specify the sequence and Tm value of PCR primers ITS-P5 and ITS-U4 used in present study.

6. Line 124-127: Please specify the method used for evaluation of effect of different drying method on the morphological characteristics of bryophytes.

7. What concentration of gel is used for the gel electrophoresis (DNA isolation and PCR products)??

Results:

1. Line 131: The ratio of the spectrophotometric absorbance of sample at 260/280 and 260/230 are generally used to determine the purity of nucleic acid contaminated by phenol or proteins and organic acids respectively. The present study includes DNA purity result based on only the 260/230 ratio.

2. S1 Table: Title should be completed, must be well understandable without the text. What is the difference between DNA purity and DNA concentration (Table 1& 2)? Please provide details of abbreviations (C, P, H, M, N, S, F) in the footnote of table.

3. Figure (1-9): Legends should be completed and well understandable without text.

4. Line 133-138 (Table 1): Spectrophotometer, nanodrops and gel electrophoretogram are used to determine the concentration of DNA. The extracted DNA having a ratio of absorbance A260/280- 1.8 and A260/230- 2.0-2.2 are considered as pure. The A260/230 values of three samples are less than 2.0. The contamination of organic acids might be present in sample. Please provide the justification of statement in your manuscript

5. Instead of long fragment DNA author can use “high-molecular weight genomic DNA”.

6. Provide details (name, collection site, specimen no. family, etc.,)in tabular form about the bryophytes specimens used in present study.

Discussion

Please cite the literature associated with problems arises during collection storage and DNA isolation from bryophyte specimens and how this study provides solutions to those problems. Please give conclusion and future prospects of your study.

Reviewer #2: The manuscript brings a simple but important technical advance in the area of molecular studies of bryophytes. Considering the findings presented here, it is possible to carry out the field collection more easily and, at the same time, guarantee the obtaining of quality DNA samples for molecular investigations.

Some minor revisions are necessary:

1- Page 5, line 102: write only "DNA extraction"

2- Page 8, lines 147 to 152: for H. calcicola, the results for hot-air drying at 40°C was not commented. The results obtained for M. polymorpha was not commented. Need to reference that the paragraph is the analysis of Fig. 2.

3- Page 8, lines 153 to 155: consider reviewing the comparison between the results for the fresh freezing control group and hot-air drying at 80°C for M. polymorpha; in table S2 it is clear that, for this species, the only treatment that showed a significant difference (p > 0.05) was drying with hot air at 150°C.

4- Page 8, lines 161 to 163: consider reviewing the comparison of the success rate of PCR amplification after hot-air drying at 80°C and the other treatments; in Fig. 4 (and supplementary figures as well) it is not possible to detect a difference between this treatment and others in some cases.

6. PLOS authors have the option to publish the peer review history of their article (what does this mean?). If published, this will include your full peer review and any attached files.

Reviewer #1: No

Reviewer #2: No

---

## [Author Response · Author response to Decision Letter 0]

14 Oct 2022

Dear Editor,

Re: Manuscript ID: PONE-D-22-14636 and Title: A comparison of drying methods on the quality for bryophyte molecular specimens collected in the field.

Thank you for your letter and the reviewers’ comments concerning our manuscript. Those comments are valuable and very helpful. We have read through the comments carefully and have made corrections. Based on the instructions provided in your letter, we will upload the files named “Response to Reviewers”, “Revised Manuscript with Track Changes”, and “Manuscript”. In the “Revised Manuscript with Track Changes”, we have modified the original text in revision mode. The responses to the reviewer’s comments are marked in red and presented in the “Response to Reviewers”. We have revised the article format according to the Journal Requirements. 

The ORCID iD of the corresponding author, Shuo Shi, is https://orcid.org/0000-0001-8428-7261. The original uncropped and unadjusted gel results images were reported in the original gel files, original gel S1-8.pdf (according to the S1-8 Figs in the file Supporting Information). 

REVIEWER 1.

Title: 1. The title of manuscript is little inappropriate and incomplete.

Introduction

The text should be more inclined towards the problems associated with the sampling and DNA isolation from bryophytes species. What are the methods traditionally used for the collection of bryophytes sample? Why different treatment is required and what are their drawbacks?

Re:

The questions raised by Reviewer1 are critical. The traditional method used for bryophyte collection in the field is natural drying treatment. However, the silica gel drying method is used primarily for collecting bryophyte molecular specimens in the field. Compared to the molecular speciments dried with silica gel in the field, it is more difficult to isolate pure DNA and PCR amplification from the herbaria specimens which dried by natural drying treatment. In the long-term experimental practice, we found that treatment methods in the field had a significant impact on the molecular specimens. Therefore, this study aimed to explore the effects of different drying methods on bryophyte specimens in the field. We have supplemented it in the preface according to your suggestions.

Materials and Methods

1. Line 76-77: What is the location or collection site of the studied bryophytes specimen?

Re:

According to your suggestion, the name, collection site, specimen no., and family of the studied bryophytes specimen have been added to the manuscript (Table 1).

2. Line 79-80: All four samples were kept fresh at the beginning of the experiment…??? Means stored in -80℃

Re:

We are so sorry for misleading your understanding because our expression is not accurate enough. In order to ensure the stability of the experimental environment and equipment, the materials were dried in the laboratory in this experiment. Before entering the laboratory, the specimens were placed in sealed plastic bags with small holes to keep the samples living. In lines 79-80, the fresh material means the living bryophyte collected from the field. We have added the sentences to “Material processing-i” of the manuscript.

3. Line 86: The collected bryophytes samples from field contains mud, dirt and microscopic organisms which may cause undesirable PCR amplification. What are the steps taken to clean the collected samples?

Re:

We are sorry for the unclear description of the collected samples cleaning method. Bryophyte cleaning steps are as follows: 1) The bryophyte was moved from the substrate, and the soil and gravel were shaken out; 2) Bryophyte specimens were rinsed in the pure water, and the sand attached in the plant were removed by tweezers. Repeat this step 2-3 times; 3) Use absorbent paper to blot the water. In the manuscript, we have added some sentences to “Material processing-i”.

This method could flush out most microscopic organisms, but not all of them. In the subsequent PCR, we used a plant-specific DNA barcode by PCR primers ITS-P5 and ITS-U4 [24]. The microorganisms could hardly be amplified by this pair of primers. To make the article more understandable, we have supplemented the primers’ peculiarities as follows: 

The PCR primers ITS-P5 “5’–3’, CCTTATCAYTTAGAGGAAGGAG” and ITS-U4 “5’–3’, RGTTTCTTTTCCTCCGCTTA” were used, which are designed for plants. The PCR amplification procedure reference to Cheng et al [24], and the annealing temperature is 55°C.

4. Line 94-100: Some parts should be reconsidered and rewritten. Experiment design is not clear from the text. From the text it appears that samples are dried in oven at 150°C, 80°C, and 40°C, respectively and then kept in a paper bag in a cool, well-ventilated place, second in a sealed plastic bag with excess silica gel, and third in a sealed plastic bag in a –80°C refrigerator and then again dried at 150°C, 80°C, and 40°C. That is bit confusing.

Re:

We are so sorry for misleading your understanding. According to your suggestion, we have rewritten “Line 94-100” as follows:

iv. In the formal experiment, there are six parts materials for the experiment. These three parts were placed in the oven electric thermostatic drying at 150°C, 80°C, and 40°C, respectively (the drying time were the same as pre-experiment). The fourth part material was kept in a paper bag in a cool, well-ventilated place. The fifth part material was collected in a sealed plastic bag with excess dry silica gel. The sixth part material was contained in a sealed plastic bag and placed in a –80°C refrigerator. After all of the samples, except the sixth ones, were dry, the follow-up experiment was performed. 

5. Line 121: Please specify the sequence and Tm value of PCR primers ITS-P5 and ITS-U4 used in present study.

Re: The Tm value of primers is the theoretical value calculated according to the sequence of primers. However, in an experiment, it is often necessary to explore the anneal temperature through the Tm value of primers. We already had the proper anneal temperature of this pair of primes for bryophytes from the preliminary experiments. According to your suggestion, we supplemented the sequence and the anneal temperature in the manuscript. As follow: “The PCR amplification procedure reference to Cheng et al. [24], and the annealing temperature is 55°C.”

6. Line 124-127: Please specify the method used for evaluation of effect of different drying method on the morphological characteristics of bryophytes.

Re:

Thanks for your suggestion. Key characters used to identify taxa (families, genera, species) and the characteristics commonly used for bryophyte morphological description were compared among the results of different treatments. The character states of traditional naturally dried specimens, were observed and compared with the ones after hot-air and silica gel dried treatments. We have rewritten it as follows: 

Both macroscopic and microscopic morphological characteristics are the important evidence for bryophyte species identification. It is unknown if the treatments used in this study affected the morphological identification of the specimen. Therefore, the overall plant morphology, leaf characteristics, and leaf transverse section characteristics of a single specimen after different drying methods were compared with those of the traditional naturally dried samples. If these characteristics were consistent, it indicated that different drying methods did not affect the morphological identification of the specimens.

7. What concentration of gel is used for the gel electrophoresis (DNA isolation and PCR products)??

Re:

The concentration of agarose gel of electrophoresis is 1%. We have added it to the manuscript.

Results:

1. Line 131: The ratio of the spectrophotometric absorbance of sample at 260/280 and 260/230 are generally used to determine the purity of nucleic acid contaminated by phenol or proteins and organic acids respectively. The present study includes DNA purity result based on only the 260/230 ratio.

Re: As the reviewer said, the ratio of the spectrophotometric absorbance of the sample at 260/280 and 260/230 are generally used to determine the purity of nucleic acid. For the 260/280 ratio, a value of ~1.8 is generally accepted as “pure” for DNA and a ratio of ~2.0 is generally accepted as “pure” for RNA. A 260/280 ratio lower than 1.8 indicates significant protein contamination. Protein contamination is usually due to inadequate DNA extraction or unstable operation. It cannot be used to assess the quality of experimental materials. Pure nucleic acid solutions typically have 260/230 values in the range of 2.0–2.2. If the ratio is lower, it may indicate the presence of contaminants such as EDTA, carbohy-drates, and phenolic compounds, all of which absorb at 230 nm. In the drying process, these compounds may be degraded and complexed with DNA, resulting in a low value of 260/230. Thus, the present study manuscript includes DNA purity results based on only the 260/230 ratio.

2. S1 Table: Title should be completed, must be well understandable without the text. What is the difference between DNA purity and DNA concentration (Table 1& 2)? Please provide details of abbreviations (C, P, H, M, N, S, F) in the footnote of table.

Re:

Thanks for your advice. Purity is to detect the contamination of other small molecules in the nucleic acid solution. Concentration is the total amount of nucleic acid in the detection solution. In order to make it easier for readers to understand, we have revised it according to your suggestions: 

We have rewritten the title of S1 Table and S2 Table, and add the abbreviations (C, P, H, M, N, S, F) in the table’s footnote. The title of S1 Table was changed to “Comparison of OD 260/230 values of the four bryophytes’ DNA after different drying treatments”. The title of S2 Table was changed to “Comparisons of extract DNA concentrations of the four bryophytes after different drying treatments”.

3. Figure (1-9): Legends should be completed and well understandable without text.

Re:

According to your suggestion, we have rewritten the legends of the figures.

4. Line 133-138 (Table 1): Spectrophotometer, nanodrops and gel electrophoretogram are used to determine the concentration of DNA. The extracted DNA having a ratio of absorbance A260/280- 1.8 and A260/230- 2.0-2.2 are considered as pure. The A260/230 values of three samples are less than 2.0. The contamination of organic acids might be present in sample. Please provide the justification of statement in your manuscript

Re:

To ensure the consistency of the experimental methods, we used mCTAB (Li et al. 2013) method to extract DNA from the four samples. Due to the significant differences in the systematic position of the four samples, the plant morphology and chemical composition were also different. For example, Marchantia polymorpha belongs to Marchantiophyta, and the plant morphology is thallus. In contrast, the other three species belong to Bryophyta, and their morphology are cormus. There are different optimal DNA extraction protocols for different samples. But we tried to use one of them to ensure experiment processing consistency. In the practical application of a large number of species in the future, it is also necessary to use one method to extract DNA for most species. Perhaps the method we used was not optimal for extracting the three mosses’ DNA. This is the reason of A260/230 values of the three samples are less than 2.0. However, the purpose of this study is not to compare different methods of DNA extraction, but to compare different effects of treatment methods for the moss specimens. For this study, the consistent extraction operation scheme can ensure the comparison of the results obtained.

5. Instead of long fragment DNA author can use “high-molecular weight genomic DNA”.

Re:

Excellent advice! We have use “high-molecular weight genomic DNA” instead of “long fragment DNA”.

6. Provide details (name, collection site, specimen no. family, etc.,)in tabular form about the bryophytes specimens used in present study.

Re: According to your suggestion, the name, collection site, specimen no., and family of the studied bryophytes specimen have been added to the manuscript. (Table 1) And as the revision for your first question of methods and materials.

Discussion

Please cite the literature associated with problems arises during collection storage and DNA isolation from bryophyte specimens and how this study provides solutions to those problems. Please give conclusion and future prospects of your study.

Re:

This study aimed to explore the effects of different drying methods on bryophyte specimens, especially in the field. The long-time preserving methods and DNA extraction methods comparations are not the questions of this study, although we compared the DNA extraction methods before deciding which one to use. 

We have added the conclusion following your suggestion, which is as follows:

It is demonstrated in this study that the hot-air drying (40–80°C) was the best method for drying bryophyte molecular specimens in the field. This method causes little damage to the DNA in bryophyte samples and is also convenient to operate. It is recommended that this method be used in the future for drying bryophyte specimens in the field.

Regarding prospects, we mentioned in the last paragraph of the discussion that the method could provide a reference scheme for animals, bacteria, lichens, algae, etc.

Thank you for your comments concerning our manuscript. Those comments are valuable and very helpful.

REVIEWER 2.

The manuscript brings a simple but important technical advance in the area of molecular studies of bryophytes. Considering the findings presented here, it is possible to carry out the field collection more easily and, at the same time, guarantee the obtaining of quality DNA samples for molecular investigations. 

Re:

Thanks for your suggestions. We have answered your questions one by one, as follows

Some minor revisions are necessary:

1- Page 5, line 102: write only “DNA extraction”

Re: Thanks for your advice. We have changed it to “DNA extraction”.

2- Page 8, lines 147 to 152: for H. calcicola, the results for hot-air drying at 40°C was not commented. The results obtained for M. polymorpha was not commented. Need to reference that the paragraph is the analysis of Fig. 2.

Re: According to your suggestion, the results for hot-air drying at 40°C for H. calcicola and the results of M. polymorpha have been supplemented to the manuscript as follows: 

For H. calcicola, the total DNA concentration obtained after hot-air drying at 40°C, 80°C, natural drying, and silica gel drying resulted in an insignificant difference (p > 0.05).

For M. polymorpha, the total DNA concentration of the five treatments had no significant difference (p > 0.05).

3- Page 8, lines 153 to 155: consider reviewing the comparison between the results for the fresh freezing control group and hot-air drying at 80°C for M. polymorpha; in table S2 it is clear that, for this species, the only treatment that showed a significant difference (p > 0.05) was drying with hotair at 150°C.

Re: According to your suggestion, we have changed the results for M. polymorpha to “For M. polymorpha, only the concentration of high-molecular weight genomic DNA obtained after 150°C is the lowest.”

4- Page 8, lines 161 to 163: consider reviewing the comparison of the success rate of PCR amplification after hot-air drying at 80°C and the other treatments; in Fig. 4 (and supplementary figures as well) it is not possible to detect a difference between this treatment and others in some cases.

Re：

Thank you for your advice. In Figure 4, there are two reasons for the little difference in amplification efficiency of the concentrated drying methods:

1) As mentioned in the discussion, the natural drying method is greatly affected by the environmental humidity, and the silica gel drying is greatly affected by the replacement frequency of operation. At the experiment’s time, the laboratory’s ambient humidity was only about 20-40%. So the specimens of the natural drying method could be dried more quickly than the spencimens in the field with high humidity. When using silica gel for drying, the frequency of material replacement in this experiment is higher than in the field of mass operation. So the natural drying rate and silica gel drying rate in this experiment are higher than in the field collection.

2) In this study, we performed PCR amplification with plant-specific primers of DNA barcode, ITS. Compared with other methods that are difficult to amplify, the advantages of this method are simple operation and high amplification efficiency.

In conclusion, although there was no significant difference in the PCR success rate of different drying methods, the DNA with a low PCR rate had been significantly damaged. Although different drying methods seem to have little difference in effect, they can reflect the good quality of DNA from the 40-80°C hot-air drying method. We have revised the article as follows based on your suggestions:

The PCR amplification for samples of five different methods and the control group was conducted (S5–S8 Figs). The statistics were obtained for assessing the success rate of PCR amplification, and the results are shown in Fig 4. For the four bryophytes, there was no statistically significant difference in PCR success rate of different drying methods. However, the amplification rate of the four samples after hot-air drying at 80°C and 40°C was higher. The amplification rate of the four samples after hot-air drying at 80°C was 100%, and the amplification rate of the three samples after hot-air drying at 40°C was 100%. The success rate of PCR amplification was slightly lower after hot-air drying at 40°C; however, it was higher than that obtained from other methods.

Your comments are valuable and very helpful to us. Thank you!

---

## [Editor Report · Decision Letter 1]

3 Nov 2022

A comparison of drying methods on the quality for bryophyte molecular specimens collected in the field

PONE-D-22-14636R1

Dear Dr. Shi,

We’re pleased to inform you that your manuscript has been judged scientifically suitable for publication and will be formally accepted for publication once it meets all outstanding technical requirements.

Kind regards,

Rosani do Carmo de Oliveira Arruda

Academic Editor

PLOS ONE

Additional Editor Comments (optional):

The manuscript contains information relevant to the study of DNA in Bryophytes and will certainly have a positive impact on research on this topic. All suggestions made by the reviewers of the manuscript were accepted and carried out by the authors.All queries were kindly explained and resolved. The methodological issues raised were resolved and also answered by the authors. Thus, I consider that the answers were sufficient, and that the article is ready to be accepted for publication.

Reviewers' comments:

The manuscript contains information relevant to the study of DNA in Bryophytes and will certainly have a positive impact on research on this topic. All suggestions made by the reviewers of the manuscript were accepted and carried out by the authors.All queries were kindly explained and resolved. The methodological issues raised were resolved and also answered by the authors. Thus, I consider that the answers were sufficient, and that the article is ready to be accepted for publication.

---

## [Editor Report · Acceptance letter]

15 Nov 2022

PONE-D-22-14636R1 

A comparison of drying methods on the quality for bryophyte molecular specimens collected in the field 

Dear Dr. Shi:

I'm pleased to inform you that your manuscript has been deemed suitable for publication in PLOS ONE. Congratulations! Your manuscript is now with our production department. 

Kind regards, 

on behalf of

Dr. Rosani do Carmo de Oliveira Arruda 

Academic Editor

PLOS ONE